# Raman Spectroscopic Analysis of Steviol Glycosides: Spectral Database and Quality Control Algorithms

**DOI:** 10.3390/foods13193068

**Published:** 2024-09-26

**Authors:** Giuseppe Pezzotti, Wenliang Zhu, Takashi Aoki, Akihiro Miyamoto, Isao Fujita, Manabu Nakagawa, Takuya Kobayashi

**Affiliations:** 1Ceramic Physics Laboratory, Kyoto Institute of Technology, Sakyo-ku, Matsugasaki, Kyoto 606-8585, Japan; wlzhu@kit.ac.jp; 2Department of Molecular Genetics, Institute of Biomedical Science, Kansai Medical University, 2-5-1 Shin-machi, Hirakata, Osaka 573-1010, Japan; 3Department of Immunology, Graduate School of Medical Science, Kyoto Prefectural University of Medicine, Kamigyo-ku, 465 Kajii-cho, Kyoto 602-8566, Japan; 4Department of Dental Medicine, Graduate School of Medical Science, Kyoto Prefectural University of Medicine, Kamigyo-ku, Kyoto 602-8566, Japan; 5Department of Orthopedic Surgery, Tokyo Medical University, 6-7-1 Nishi-Shinjuku, Shinjuku-ku, Tokyo 160-0023, Japan; 6Department of Applied Science and Technology, Politecnico di Torino, Corso Duca Degli Abruzzi 24, 10129 Torino, Italy; 7Department of Molecular Science and Nanosystems, Ca’ Foscari University of Venice, Via Torino 155, 30172 Venice, Italy; 8Faculty of Fiber Science and Engineering, Kyoto Institute of Technology, Sakyo-ku, Matsugasaki, Kyoto 606-8585, Japan; t-aoki@kit.ac.jp; 9Morita Kagaku Kogyo Co., Ltd., 1-19-18 Inadaue-machi, Higashi Osaka, Osaka 577-0002, Japan; miyamoto-akihiro@morita-kagaku-kogyo.co.jp (A.M.); fujita-isao@morita-kagaku-kogyo.co.jp (I.F.); 10Department of Medical Chemistry, Kansai Medical University, 2-5-1 Shin-machi, Hirakata, Osaka 573-1010, Japan; nakagawc@hirakata.kmu.ac.jp (M.N.); kobayatk@hirakata.kmu.ac.jp (T.K.)

**Keywords:** steviol glycosides, Raman spectroscopy, quality control algorithms, Raman database

## Abstract

Besides all sharing an extraordinary high (i.e., up to ~450 times) sweetening power as compared to sucrose and while presenting strong similarities in their molecular structures, molecules belonging to the family of diterpene glycosides (i.e., the secondary metabolites of *Stevia rebaudiana*) differ in specific structural details that strongly impact on their levels of sweetness and bitter aftertaste. Given the nutritional and pharmacological benefits of steviol secondary metabolites as natural dietetic and anti-diabetic remedies, extraction and purification of steviol glycosides from plant material are nowadays widely spread among many countries. However, an unpleasant bitter aftertaste, which is linked to a genetic variation in human bitter taste receptors, hampers the full exploitation of such benefits and calls for a prompt improvement in organoleptic property control of stevia products. A deeper understanding of the molecular structure of different steviol glycosides and the consequent development of promptly measurable criteria for the organoleptic performance of their mixtures will support processing optimization and control of taste profiles within desired yields. The present research aimed at establishing Raman spectroscopic algorithms for quantitative characterizations of raw stevia-based sweetener products. First, a series of twelve high-purity diterpene glycosides were analyzed by high spectrally resolved Raman spectroscopy and their spectra analyzed in order to establish a complete Raman library of molecular structures. Then, quantitative spectroscopic parameters were built up and applied to characterize the organoleptic property of five different commercially available samples including the recently developed Rebaudioside M isoform. Raman spectroscopy was confirmed as a versatile analytical technique that could be used for quantitative quality control tasks on the production line and for prompt in situ characterizations of purchased products.

## 1. Introduction

Diterpene (steviol) glycosides, namely, the secondary metabolites derived from the leaves of the *Stevia rebaudiana* plant, are nowadays widely used as natural sweeteners [1,2]. Known for their intense sweetness without the added calories or carbohydrates found in traditional sugars, these natural molecules have gained popularity as sugar substitutes in various parts of the world. The sweet taste of steviol glycosides is 150~450 times more intense than sucrose [3], making it a desirable option for those looking to reduce their sugar intake or manage pathological conditions like diabetes. One of the significant advantages of steviol-containing sweeteners is that they are natural, plant-based, and safe alternatives to artificial sweeteners [3,4,5]. Using them as sugar substitutes in food and beverages, including soft drinks, desserts, and various processed products, could represent a valid alternative to strict diets and the associated risks of metabolic damage and muscle loss [6]. Steviol-containing sweeteners, such as Stevioside and Rebaudioside A, are generally recognized as safe by regulatory authorities in many countries [7]. Stevia-based sweeteners are valued for their potential to provide sweetness without contributing to the rise in blood sugar levels, making them a popular choice for individuals with diabetes. Overall, steviol-containing sweeteners offer a natural and low-calorie alternative to traditional sugars, making them a popular choice for individuals seeking sweeteners with fewer calories and potential health benefits.

Despite the above indisputable advantages, a large fraction of people perceive a bitter aftertaste with steviol glycoside-based sweeteners, such undesired taste varying among different brands and formulations [8]. Different rebaudiosides, despite all being secondary metabolite isomers from the leaves of the *Stevia rebaudiana* plant, could have markedly different aftertastes. All rebaudiosides contribute to the overall sweetness of the plant, but variations in their chemical structures result in differences in taste profiles. The two so far more widely studied rebaudiosides are Rebaudioside A and Rebaudioside C. The former is often considered to have a more sucrose-like (or sugar-like) taste; scoring 200~400 times sweeter than sucrose, it is characterized by a clean, sweet flavor without significant bitterness. On the other hand, the latter provides sweetness with lower potency while also having a more bitter or licorice-like taste as compared to the former. Depending on relative fractions, they thus contribute different flavor profiles to stevia extracts. It should also be noted that a number (twelve so far) of different diterpene glycosides have been discovered with different taste characteristics, and stevia extracts used in commercial products often contain variable combinations of them. Therefore, any specific blend links to a different taste. The relative proportions of different glycoside isomers can vary depending on factors such as plant source, extraction method, and adopted purification processes. Given also that the perception of taste is subjective with some people being more sensitive to certain taste characteristics, stevia sweeteners continue to be developed and refined. Researchers and manufacturers are exploring different blends of rebaudioside isomers and their combinations to achieve optimal taste profiles with minimal bitter aftertaste [8,9,10,11,12,13,14,15]. The most recent development is Rebaudioside M [16,17], which is valued for its superior sweetness and clean taste. This newly discovered species is known for being so far the most sugar-like tasting rebaudiosides, its sweetness intensity being comparable to or even greater than Rebaudioside A with less bitterness or licorice-like aftertaste compared to any other known rebaudiosides. Its taste profile is thus particularly appealing for those seeking a closer match to the taste of sucrose.

Several analytical methods have been used to characterize and quantify steviol glycosides. These methods have played a key role in quality control, research, and regulatory compliance in the production of stevia-based sweeteners. High-performance liquid chromatography and its combination with mass spectrometry have been used to separate, identify, and quantify with high sensitivity and specificity the individual steviol glycosides present in mixtures [18,19,20]. Nuclear magnetic resonance spectroscopy [21,22], capillary electrophoresis [23,24], enzyme-linked immunosorbent assay [25], and ion chromatography [26] have also been used for analyzing ionic compounds in stevia extracts. Although the choice of method depends on factors such as the specific glycosides of interest, the required sensitivity, and the equipment available in the laboratory, all the above methods are expensive, time consuming, and require complex sample preparation.

Vibrational spectroscopies represent a valid alternative in characterizing stevia products since they are capable of identifying specific functional groups within the molecules of steviol glycosides [27,28,29]. This information aids in distinguishing characteristic chemical bonds in both steviol aglycone and sugar moieties, and enables analyses of the arrangement of both sugar units and aglycone core structure, thus ultimately quantifying the amounts of different steviol glycosides in the sample. Raman spectroscopic measurements can be performed directly on samples without the need for extensive sample preparation. This feature is also advantageous for studying steviol glycosides in their natural environment and their possible counterfeits [30]. However, Raman spectroscopy of steviol glycosides yet appears in its infancy, as a complete library of elementary molecules is not yet available and spectroscopic algorithms for quantitative analyses are conspicuously lacking. The present study aims at filling these gaps by delving into the spectroscopic details of these complex molecules. In particular, we provide a complete series of high-resolution and statistically validated Raman spectra of all the twelve so far known diterpene glycosides, while also proposing quantitative spectroscopic criteria to judge about the degree of sweetness and bitter retrotaste of mixtures present in commercial stevia products.

## 2. Materials and Methods

### 2.1. Samples and Their Preparation Procedures

The compounds studied, which are listed in Table 1 together with their chemical characteristics, were all purchased from ChromaDex, Inc. (Los Angeles, CA, USA). Five commercially available Stevia samples, referred to as China 1, China 2, Morita 1 (Morita Chemical Co., Ltd., Osaka, Japan), Morita 2 (Morita Chemical Co., Ltd., Osaka, Japan), and Fermented, were purchased in Japan.

### 2.2. Raman Spectroscopy

The Raman spectral library of the elementary diterpene glycoside molecules and the spectra of commercially available products were all compiled by means of a high spectrally resolved Raman device equipped with a triple-monochromator (T-64000; Jobin-Ivon/Horiba Group, Kyoto, Japan) and a nitrogen-cooled charge-coupled device detector (CCD-3500V, Jobin-Ivon/Horiba Group, Kyoto, Japan). The excitation source was the 514 nm line of an Ar-ion laser operating with a nominal power of 200 mW. The spectral resolution was better than 0.5 cm^−1^. Series of 10 spectra were collected (with a 50× optical lens) at randomly selected locations of each sample and then averaged to obtain a representative spectrum per each compound or product. The collected Raman spectra were subjected to polynomial baseline subtraction and deconvolution into a series of Gaussian-Lorentzian band components. The baseline subtraction procedure was performed using options available in commercial software (LabSpec 4.02, Horiba/Jobin-Yvon, Kyoto, Japan) with fixed criteria for all collected spectra. All spectra were analyzed after intensity normalization to the strongest signal in the collected spectral interval. Based on selected molecular structure characteristics and their related spectroscopic signals, three spectroscopic parameters were located, which related to the number of glucose rings and carbonyl bonds. These parameters reflected the degree of sweetness and bitter retrotaste. Calibration curves linking these spectroscopic parameters to the structure of specific diterpene molecules were then built up upon least-square fitting of data collected on the full series of elementary diterpene molecules investigated.

### 2.3. Evaluation of Sweetness Quality

Each purified product of steviol glycoside and commercial product was dissolved in water, and an aqueous solution containing 0.05 wt.% of each product was prepared as a sample. Sensory evaluation was conducted by 10 panelists experienced in sensory testing of steviol glycosides. Panelists were not informed in advance which rebaudioside or commercial product, other than rebaudioside A, corresponded to the sample placed in the container. Each panelist scored each sample according to a scale including sweetness (high/low), sweet perception rate (fast/slow), and bitter perception rate (fast/slow).

## 3. Experimental Results

### 3.1. Raman Spectroscopic Library of Elementary Compounds

Figure 1 shows schematic drafts of the molecular structures and the respective high spectrally resolved Raman spectra of (a) Dulcoside A, (b) Stevioside, (c) Steviolbioside, and (d) Rubusoside. Similarly, Figure 2a–h show molecular structures and high-resolution Raman spectra of Rebaudiosides A, B, C, D, E, F, M, and N, respectively. As seen at a first glance, all 12 spectra in Figs. 1 and 2 present strong similarities to each other, but also clear differences in both spectral shifts and relative intensities. Such details, which arise from different structural features at the molecular scale, indeed represent precious hints to be interpreted and used to spectroscopically distinguish among different diterpene glycoside molecules. Figure 3 and Figure 4 give tentative spectral regions and wavenumbers for the most significant vibrational contributions in α–D–glucose ring and steviol aglycone core, respectively. Six main regions were located for vibrational modes intrinsic to the glucose rings (cf. labels in inset to Figure 3). From low to high wavenumbers, such regions represent bending modes of core pyranose ring bonds, stretching modes of core pyranose ring bonds, and bending modes of functional groups at C1~C4 and the hydroxymethyl group linked to C5. Main aglycone core vibrations in Figure 4 (cf. labels in inset) display six main bands, as follows: O–C19=O bending at 740 cm^−1^ (strongest in intensity in Steviolbioside lacking any glucose ring at C19 side; cf. Figure 1c), C11–C12 stretching at 898 cm^−1^, C–C bond stretching in aglycone rings in the wavenumber interval 1004~1075 cm^−1^, C9–C11–H stretching at 1204 cm^−1^, C16–C17 stretching at 1670 cm^−1^, and C19=O carbonyl stretching at ~1738 cm^−1^. An additional carbonyl-stretching mode is contemplated in Figure 4h, in which the C17 side of the molecule undergoes a structural modification in presence of oxygen and water molecules to incorporate an additional carbonyl unit at C16. This possible configuration was suggested by the presence of an additional carbonyl sub-band at ~1706 cm^−1^ in the spectrum of the pure Steviolbioside compound (cf. Figure 1c).

The possible existence of this modification at the C17 site cannot completely be ruled out in other steviol molecules although the band at 1706 cm^−1^ could not be clearly resolved. With reference to Figure 3 and Figure 4, asterisks in inset to Figure 1 and Figure 2 show the main Raman vibrations related to aglycone core that were selected for the present analysis. On the other hand, spectral areas located with broken-line squares represent areas for the Raman modes of glucose rings selected in the present analysis, as follows: C–O and C–C stretching at 850~950 cm^−1^, and C–C–H and C–O–H bending at 1150~1250 cm^−1^. Selected spectroscopic parameters based on the above analyses are given in the next section.

### 3.2. Raman Parameters for Structural Assessments of Steviol Diterpenes

Figure 5a–d show schematic drafts of the C19 side of diterpene molecules with 0, 1, 2, and 3 glucose rings at the C19 side, respectively. We hypothesized that the wavenumber of the carbonyl C19=O stretching sub-band will increase with increasing the number of rings, the higher the number of rings the higher the energy required for such vibrational mode to occur. Highly resolved and deconvoluted experimental Raman spectra in the wavenumber region 1600~1800 cm^−1^ are shown in Figure 5e–h for all 12 studied (pure) diterpene molecules with 0, 1, 2, and 3 glucose rings at C19 side, respectively. As seen, an increased wavenumber trend was indeed found with increasing the number of rings at their C19 side, with the anomaly of the Steviolbioside molecule, which, despite having 0 rings at its C19 side showed two C=O bands, at 1706 and 1738 cm^−1^. This anomaly was interpreted with hypothesizing a modification of the C17 site to incorporate an additional carbonyl group (cf. Figure 4h), as proposed in the previous section. Plots of carbonyl-stretching wavenumbers for 12 elementary molecules, as a function of C19-side ring numbers, are shown in Figure 6a,b. Despite slight variations, in particular for molecules with 1 and 3 rings, which were obviously due to the stereometric details of ring-aglycone and inter-ring bonds, a monotonically increasing fitting curve could be obtained as a function of ring number, as shown in Figure 6b. In order to evaluate the total number of rings linked to the 12 elementary diterpene molecules studied, two additional parameters were proposed. The first parameter exploits the relative intensity ratio between the C5–O5 stretching band in glucose rings (at 887 cm^−1^; Figure 7a) and the C11–C12 stretching sub-band of the aglycone core (at 898 cm^−1^; cf. Figure 7b); the (areal) intensity ratio of these two signals is referred to as *R*_1_ = *I*_887/I898_, the higher the *R*_1_ value the higher the number of rings per unitary molecule. In Figure 7c–g, sub-bands at 887 and 898 cm^−1^ (as extracted from the deconvoluted spectra in Figure 1 and Figure 2) are replotted for elementary molecules with cumulative numbers of rings on C13/C19 sides equal to 2, 3, 4, 5, and 6, respectively. The computed *R*_1_ values, which are given in inset to each figure, appear to obey an increasing trend with increasing the total number of rings. A similar procedure was followed upon selecting C–O–H bending modes in glucose rings (at 1192 and 1215 cm^−1^; Figure 8a) and the C11–C9H stretching mode in the aglycone core (at 1204 cm^−1^; Figure 8b). This latter spectroscopic approach led to define an additional Raman ratio as: *R_2_* = (*I*_1192_ + *I*_1215_)/*I*_1204_. In Figure 8c–g, sub-bands at 1192, 1215, and 1204 cm^−1^ (as extracted from the deconvoluted spectra in Figure 1 and Figure 2) are replotted for elementary molecules with cumulative numbers of rings on C13/C19 sides equal to 2, 3, 4, 5, and 6, respectively. Similar to the case of the *R*_1_ ratio, the computed *R*_2_ values (given in inset to each figure) appear to obey an increasing trend with increasing the total number of rings. Intervals for the detected values of *R*_1_ and *R*_2_ ratios are plotted and compared in Figure 9a, while plots of the *R*_1_ and *R*_2_ values computed for 12 elementary molecules, as a function of the cumulative ring numbers on C13/C19 sides ring numbers are shown in Figure 9c, respectively (cf. labels in inset). Despite differences among elementary molecules in stereometric details of ring-aglycone and inter-ring bonds, monotonically increasing fitting curves could be obtained with relatively small scatters as a function of ring number. The fact that two independent spectroscopic parameters appear capable to identify the total number of glucose rings in elementary diterpene molecules, strengthens the idea that Raman spectroscopy could be capable to analyze the structure of stevia products.

### 3.3. Evaluation of Commercially Available Stevia Products

Figure 10a–e show high spectrally resolved and deconvoluted Raman spectra for the commercially available stevia samples referred to as China 1, China 2, Morita 1, Morita 2, and Fermented, respectively (cf. Section 2.1). In the respective spectra, similar to Figure 1 and Figure 2 and with reference to Figure 3 and Figure 4, asterisks and broken-line squares in inset locates the main Raman vibrations related to aglycone core and selected spectral areas for Raman modes of glucose rings, respectively. In order to better visualize the three spectroscopic parameters, as defined in the previous section, namely, carbonyl wavenumbers, and intensity ratios *R*_1_ and *R*_2_, deconvoluted sub-bands in spectral intervals at 1700~1750 cm^−1^, 850~950 cm^−1^, and 1175~1225 cm^−1^, were replotted in Figure 11a–c, respectively. Wavenumbers for the deconvoluted bands and average values for the above three spectroscopic parameters are given in inset to each figure. The letters B and M/N in Figure 11a correspond to 0 and 3 ringsides (i.e., Rebaudioside B and M and/or N, respectively). In the same figure, percent are fractions given in inset, which refer to the areal percent covered by each carbonyl sub-band over the total area of the sub-bands in the spectral zone of carbonyl stretching. Since the spectral locations of the carbonyl stretching sub-bands are related to the molecular structure of specific steviol glycosides (as explained in Section 3.1), the shown sub-band fractional areas reflect in a semi-quantitative ways the fractions of different steviol glycosides contained in the blend of each commercial product.

Based on the average values of the three spectroscopic parameters computed, the respective taste sensorial performances of the five commercial stevia products were matched with the calibration curves obtained from elementary molecules (cf. Figure 6 and Figure 9). These comparisons are shown in Figure 12a–c for carbonyl wavenumbers, and intensity ratios *R*_1_ and *R*_2_, respectively. Increasing values on each plot were considered as measures of increased sweetness and lowered bitterness, as discussed in detail in the discussion sections in the reminder of this paper. It is immediately clear that the five examined commercially available stevia products contained quite different blends of diterpene molecules, as a specific characteristic of both quality of original stevia leaves and steviol extraction procedures. Table 2 summarizes all three Raman spectroscopic parameters for the five examined commercial products and, in addition, shows the computation of the number of rings on C13 side, as obtained from the subtraction of the number of rings on C19 side from the total number of rings. In Table 2, a qualitative summary of taste characteristics is also given, as discussed in the next Section 4.1 and Section 4.2. Finally, an important spectroscopic detail, which only appeared in the samples China 2 and Fermented, was the presence of a band at ~869 cm^−1^ (labeled with an arrow and an asterisk in Figure 11b). This relatively weak band appears to be extraneous to any vibrational mode of steviol diterpenes and could be hypothesized as the spectroscopic fingerprint for a small amount of added artificial sweetener (likely sodium cyclamate) [30]. We looked for additional bands from artificial sweeteners in order to confirm the above hypothesis. However, additional bands from artificial compounds resulted conspicuously overlapped to diterpene bands and could be hardly singled out. Accordingly, this hypothesis will need additional confirmation in future studies.

## 4. Discussion

### 4.1. The Molecular Origin of Sweetness and Bitter Retrotaste

Since the early study by Kubota and Kubo [31], the chemical basis for the perception of bitter aftertaste, as generally associated with diterpene glycosides, has been elucidated as stemming from the presence of three characteristic attachment loci, namely, a proton donor group, a proton acceptor group, and a third binding site that interacts with the hydrophobic site on the receptor. While the perception of bitterness is known to vary among different individuals due to genetic factors [32], it is nowadays well established that the steviol aglycone core, in association with the specific molecular arrangement of D-glucose rings on both its C19 and C13 sides, is responsible for the bitter aftertaste upon its interaction with bitter taste receptors [10,33,34,35]. The spatial arrangement and the interactions among different functional groups in diterpene molecules greatly influence the overall three-dimensional structure. Accordingly, steric effects, namely, the spatial hindrance caused by adjacent atoms or groups, clearly impact on binding of the glycoside to taste receptors. According to Acevedo et al. [36], the interactions between steviol glucosides and the binding pocket of bitter taste receptors are governed by both hydrogen bonding and hydrophobic interactions: oxygen and nitrogen atoms from amino acid residues of the binding pocket interacting with the OH groups of monosaccharides bound at positions C13 and C19 through hydrogen bonds, while the carbons of the aglycone part interacting with apolar residues at the binding site through hydrophobic interactions. Advances in research, including studies using sensory analysis and molecular modeling, have given insights into the specific factors influencing bitterness perception in steviol glycosides [36,37,38,39]. Bitterness intensity was found to basically depend on the two following molecular-scale circumstances: (a) diterpene glycoside molecules with more sugars have less affinity towards bitter taste receptors, because the size of the receptor binding cavity is limited and cannot accommodate multiple glucose units; and (b) the presence of transmembrane regions in the binding cavity of specific receptors with polar amino acid residues favors hydrophobic contact with carbon sites of the aglycone core.

As widely recognized regarding sweetness [37], the key chemical circumstance from the molecular viewpoint is the formation of hydrogen bonds with receptor residues. Since this is regulated by multipoint stimulation [40], it is obvious that steviol glycoside molecules with more glucose residues will have an increased probability to form hydrogen bonds with receptor sites, thus releasing stronger sweetness than glycoside molecules with fewer substituent moieties [41]. Moreover, as a consequence of multipoint stimulation, a higher binding efficiency is expected in maximizing sweetness due to the higher number of compound/receptor binding sites [42]. 

The above notions justify the findings that Rebaudioside M, Rebaudioside D, and Rebaudioside A with more glucose residues (6, 5, and 4 rings, respectively) possess higher maximum sweetness and faster sweetness increase rates as compared to Rebaudioside C, Stevioside, and Rubusoside (3, 3, and 2 rings, respectively) [33]. On the other hand, the level of sweetness in Rebaudioside C and Rebaudioside A differs because of their different substituents on C13, namely, rhamnose and glucose, respectively (cf. Figure 2a,c, respectively). Compared to glucose, rhamnose is missing one hydroxyl group, thus reducing the number of hydrogen bonds in Rebaudioside C as potential hydrogen links with taste receptors and, consequently, leading to weaker sweetness and lower sweetness increasing rate.

It appears clear from this latter case that the level of sweetness in steviol glycosides is not always only related to the total number of glucose rings per unit volume, but also to the molecular details of specific structural features.

According to Tao and Cho [43], the bitterness of Rebaudioside A stood out when consumers first tasted the sample, while continuing even after one minute. The bitterness of Rebaudioside A was indeed reported by several other researchers [10,44]. On the other hand, Rebaudiosides D and M were widely recognized as possessing quite low long-term in-mouth bitterness and a sweetness with an intensity similar to sucrose. However, unlike Rebaudioside M, Rebaudioside D exhibited a significantly higher immediate bitterness than sucrose [16]. Two distinct research groups, namely, Hellfritsch et al. [10] and Ko et al. [45], have reported that Rebaudioside D elicited significantly less bitterness as compared to Rebaudioside A. 

In agreement with the above results, time-intensity dynamic sensory characterizations of temporal sweet and bitter perceptions of Rubusoside, Stevioside, Rebaudioside C, Rebaudioside A, Rebaudioside D and Rebaudioside M confirmed that Rebaudioside M and Rebaudioside D possess faster onset of sweetness, quicker decay of bitter aftertaste, while being almost completely devoid of bitterness [33]. On the other hand, Rubusoside and Stevioside were characterized by an immediate and distinct bitter taste and a lingering bitter aftertaste. According to Tian et al. [33], phenomenological data pointed to adsorption/desorption characteristics on taste receptors, as follows:(i)Fewer glucosyl groups on C19 result in shorter time for initial stimulation and longer perception of bitterness.(ii)The more the glucosyl groups on C13 the faster the increase and the stronger the intensity of sweetness.(iii)A lower ratio between the number of glucosyl groups on C13 to that on C19 leads to a faster sweetness peak perception, although this parameter did not affect the bitter taste.(iv)Higher numbers and larger sizes of substitutions at the C19 position of steviol glycosides increase desorption and lead to a quicker decay of sweetness.(v)Rubusoside and Stevioside compounds, which contain fewer glucosyl groups, undergo lower desorption and thus longer bitter aftertaste.(vi)The addition of glucosyl groups tends to concurrently generate stronger sweetness and less bitterness, but only when the number of substituents on C13 is close to that on C19.

In the next section, we shall attempt to propose quantitative Raman algorithms capable to give objective measurements of sweetness vs. bitter aftertaste in the 12 analyzed steviol glycosides.

### 4.2. Raman Algorithms to Assess Sweetness/Bitterness of Commercial Stevia

Researchers and food scientists continue to explore strategies to optimize the taste profile of stevia sweeteners towards the development of steviol glycosides with lowest/shortest bitter and highest/fastest sweet perceptions. Concurrently, a number of different analytical techniques have been proposed in support of those strategies for quantitative and objective characterizations of taste perception in stevia products [19,20,21,22,23,24,25,26,27,28,29,30]. The originality of the present Raman studies resides in having built up for the first time a precise spectral library of all known elementary steviol diterpenes, from which spectroscopic criteria for taste perception could be extracted upon analyzing monitoring key spectroscopic signals. The Raman analyses enabled assessing average estimates for the number of glucose rings on the C19 side, the total number of rings (based on two distinct algorithms), and, consequently, the C13 to C19 ratio of the numbers of rings at both sides of the steviol aglycone core. Table 2 lists the above parameters for the five commercial products investigated (cf. Section 2.1). According to the Raman data and following the time-intensity dynamic sensory characterizations of temporal sweet/bitter perceptions analyses, as given in Ref. [33] (cf. Section 4.1), the sample labeled as Morita 1 showed the highest total number of rings (cf. consistent values by two different spectroscopic approaches in Table 2), the highest number of rings on the C19 side and the lowest C13/C19 ring ratio among all tested stevia samples. According to these spectroscopic characteristics, the Morita 1 sample possessed the highest level of sweetness, the fastest sweet and the shortest bitter perceptions. On the other hand, the samples labeled as China 2 and Fermented showed exactly the opposite characteristics, thus intrinsically being the poorest products from the viewpoint of steviol molecular structure. Note, however, that both these products showed traces of artificial sweeteners (cf. green asterisks in Figure 11b). Apparently, the respective makers have accomplished with the need to adjust the poor intrinsic diterpene characteristics through adding small doses of synthetic sweetener molecules. The remaining two samples, namely, Morita 2 and China 1, did not appear to contain artificial sweeteners, but displayed relatively poor molecular structures in terms of both level of sweetness and bitter retrotaste (cf. their quite low number of rings on C19 side and very high C13/C19 number of ring ratio in Table 2).

The sample Morita 1, which appeared as the one with the best taste sensorial properties, was indeed rich in Rebaudioside M. Although specific structural characteristics of Rebaudioside M resulting in decreased affinity for bitter taste receptors as well as the complexity of such molecular interactions are not yet fully understood, some general principles and hypotheses based on the structure-activity relationship of steviol glycosides provide insights: (a) the Rebaudioside M molecule is the richest in sugar moieties (cf. Figure 2g), whose specific arrangement and position play a crucial role; and, (b) the spatial arrangement of the sugar units on the Rebaudioside M molecule may create steric hindrance, which in turn could affect the glycoside’s ability to bind or interact with bitter taste receptors. The affinity for bitter taste receptor-binding sites sharply decreases as a consequence of any molecular arrangement unfavorable to their engagement. Therefore, conformational changes in the molecule can influence how it fits into the binding sites on taste receptors, having a favorable interaction with sweet taste receptors (T1R2/T1R3) [34], while experiencing a reduced interaction with bitter taste receptors. Note, however, that stevia developers have encountered difficulties in obtaining Rebaudioside M from the natural plant source, because of its relatively low abundance as compared to other steviol glycosides, (e.g., Rebaudioside A). In summary, Rebaudioside M is highly prized for its clean sugar-like taste with minimal bitterness or aftertaste, but its scarcity in the natural plant continues to pose challenges in its commercial production. The Morita 1 products appears uniquely capable to include a considerable fraction of Rebaudioside M, which leads to its superior structural characteristics as compared to other commercial products, as shown in Table 2. 

As shown in Figure 5 and Figure 6, the wavenumber of carbonyl stretching in the aglycone core shifts toward higher wavenumbers with increasing the number of glucose rings on C19 side. Accordingly, Steviolbioside and Rebaudioside B (with zero rings at C19) both show carbonyl bands at the lowest wavenumber of 1706 cm^−1^. Spectral deconvolution in the C=O zone between 1700 and 1750 cm^−1^ (cf. Figure 11a) showed that Morita 1 and Morita 2 were the stevia products with the lowest and the highest spectral intensities (1.9 vs. 18.1%) at <1710 cm^−1^, respectively. None of these two products shows signals at around 1738 cm^−1^. In other words, the amount of molecules with zero rings are minimal in Morita 1 and maximum in Morita 2, while both products were missing C=O linked zero C19-ring molecules (cf. Figure 4g). China 1 and China 2 also present relatively low fractions (4.2%) of carbonyl spectral intensity at ~1705 cm^−1^. However, they also present relatively strong signals (i.e., 34.3 and 21.4% of the overall C=O Raman activity) at 1736~1738 cm^−1^, which suggest a prominent presence of linked carbonyl sites (cf. Figure 4h). A low sensorial performance could thus be expected for both these products due to the prominent presence of linked Steviolbioside. Moreover, China 2 also shows a relatively low amount (~21.0%) of Rebaudioside A, with signal at ~1719 cm^−1^. The above two spectral characteristics in China 1 and China 2 products are reflected in their relatively low total number of rings (cf. Figure 12). Also the Fermented product shows a C=O spectroscopic pattern leading to a quite poor sensorial performance, with relatively strong carbonyl bands at 1702 and 1738 cm^−1^, and low signals at higher wave-numbers related to Rebaudioside A, C, and F. The present Raman data, while confirming that the taste sensorial performance of stevia products is a complex result of multiple factors related to each specific blend of molecules, also clearly show that it is possible to judge about the sensorial taste performance of commercially available stevia products by means of Raman spectroscopic assessments. 

### 4.3. The Role of Raman Spectroscopy in Quality Control of Stevia Products

Based on the evidences given in this paper, Raman spectroscopy can play a significant role in the quality control of stevia products, particularly in assessing the chemical composition and structural characteristics of steviol glycosides, the compounds responsible for the sweet taste in stevia. Besides ensuring that the product contains the desired sweetening compounds and confirming the absence of undesired components, we show here that Raman spectroscopy can be used for quantitative analysis, allowing for an estimation of the concentration of specific steviol glycosides in a sample. The presented Raman parameters are important for quality control to ensure that the product meets regulatory standards and consumer expectations. Without the need for any extensive sample preparation, the analysis of steviol Raman spectra can also be valuable, as also shown in this paper, in detecting any variations or contaminants that may affect quality (i.e., impurities, adulterants, and artificial sweeteners that might be present in commercial products) [30,46,47,48,49,50]. This is crucial for ensuring the purity of the sweetener and its compliance with food safety standards. Moreover, Raman spectroscopy could be applied to non-destructively monitor the manufacturing processes involved in the production of stevia products. Finally, upon assessing the chemical changes during successive processing steps, it can help optimize and control the production parameters to achieve the desired product quality. 

## 5. Conclusions

In the present study, we have shown that Raman spectroscopy can quantitatively be used to identify and count specific functional groups within the molecules of steviol glycosides. This information aids in confirming the amounts of characteristic chemical bonds, such as those found in steviol aglycone and sugar moieties. We provided for the first time a complete high-resolution library of Raman spectra for all the known elementary diterpene glycoside molecules, which could serve as references to Raman spectroscopists interested in analyzing stevia products. In addition, Raman spectroscopic parameters have also been proposed, which were tailored according to the time-intensity dynamic sensory characterizations of temporal sweet/bitter perceptions analyses applied here as reported by other authors. The proposed parameters could be used for correlating relative intensities and spectral shifts of selected Raman bands with the concentration of specific compounds. This approach opens the way to quantify the amounts of different steviol glycosides in stevia samples through in situ real-time analyses, without the need for any extensive sample preparation or expensive/time-consuming analyses. Finally, it is foreseen that the analytical characteristics of Raman spectroscopy could be particularly advantageous in studying steviol glycosides in their natural environment and in monitoring their modifications during processing and manufacturing (i.e., including chemical changes, stability, and the impact of processing conditions).

## Figures and Tables

**Figure 1 foods-13-03068-f001:**
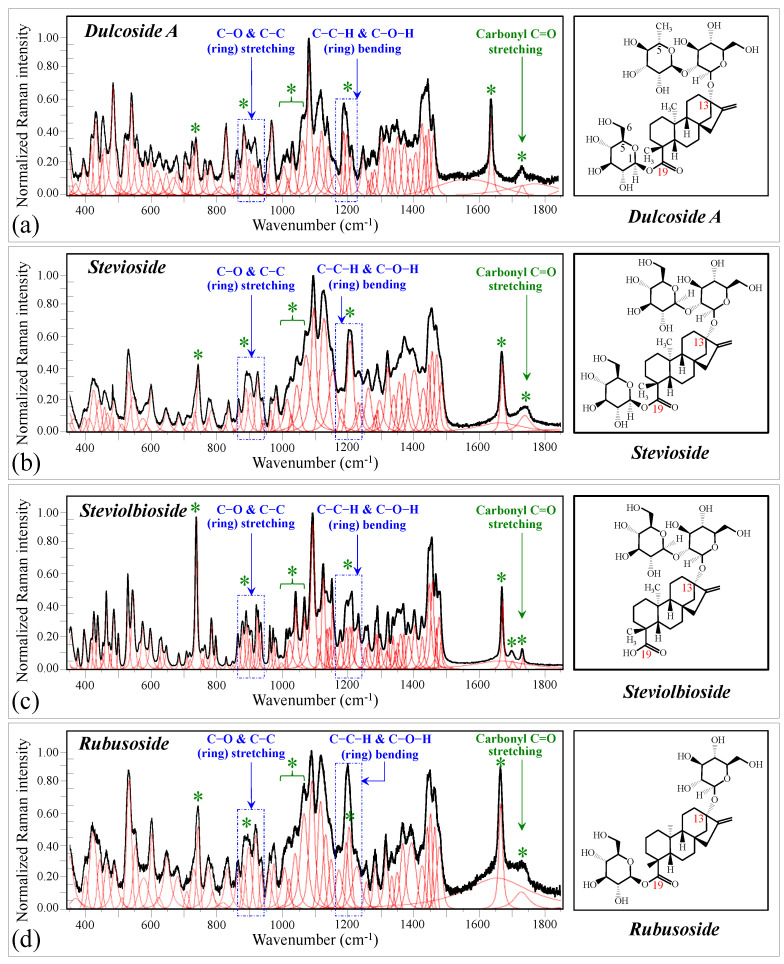
Deconvoluted Raman spectra (black and red lines for spectra and deconvoluted sub-bands, respectively) and molecular structures of four distinct diterpene steviosides: (**a**) Dulcoside A, (**b**) Stevioside, (**c**) Steviolbioside, and (**d**) Rubusoside. Wavenumber ranges including the specific molecular vibrations from glucose rings, as employed in this study for building up spectroscopic algorithms of Raman molecular evaluation, are shown in inset; green asterisks represent spectral band components specifically assigned to the aglycone core (cf. text).

**Figure 2 foods-13-03068-f002:**
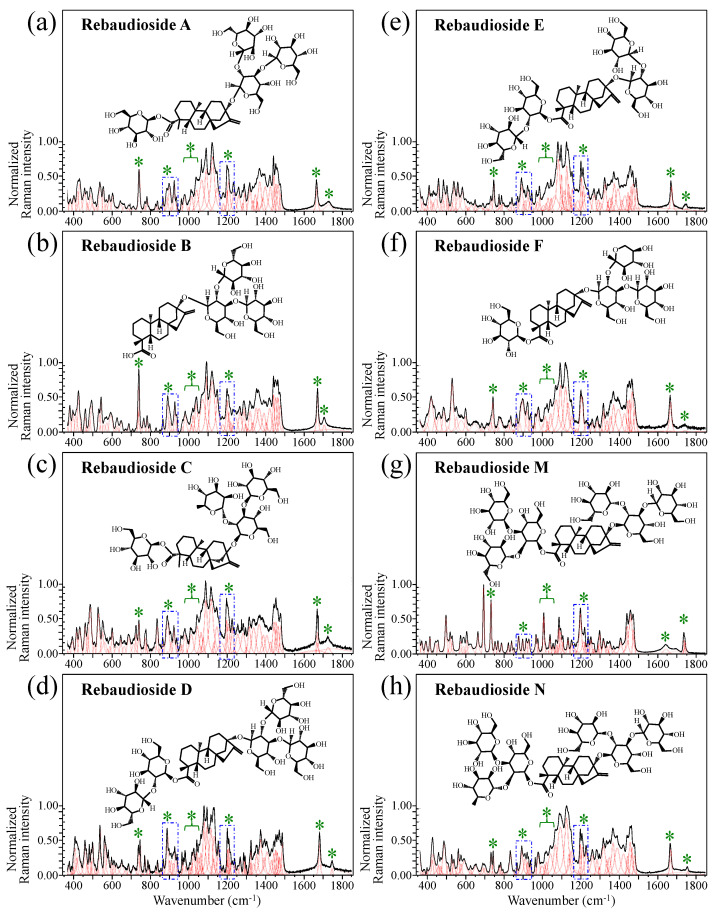
Deconvoluted Raman spectra (black and red lines for spectra and deconvoluted sub-bands, respectively) and molecular structures of eight distinct rebaudioside molecules: (**a**) Rebaudioside A, (**b**) Rebaudioside B, (**c**) Rebaudioside C, (**d**) Rebaudioside D, (**e**) Rebaudioside E, (**f**) Rebaudioside F, (**g**) Rebaudioside M and (**h**) Rebaudioside N. Wavenumber intervals including the specific molecular vibrations from glucose rings, as employed in this study for building up spectroscopic algorithms of Raman molecular evaluation, are shown with broken-line squares (same as those shown in Figure 1); green asterisks represent spectral band components specifically assigned to the aglycone core (cf. text).

**Figure 3 foods-13-03068-f003:**
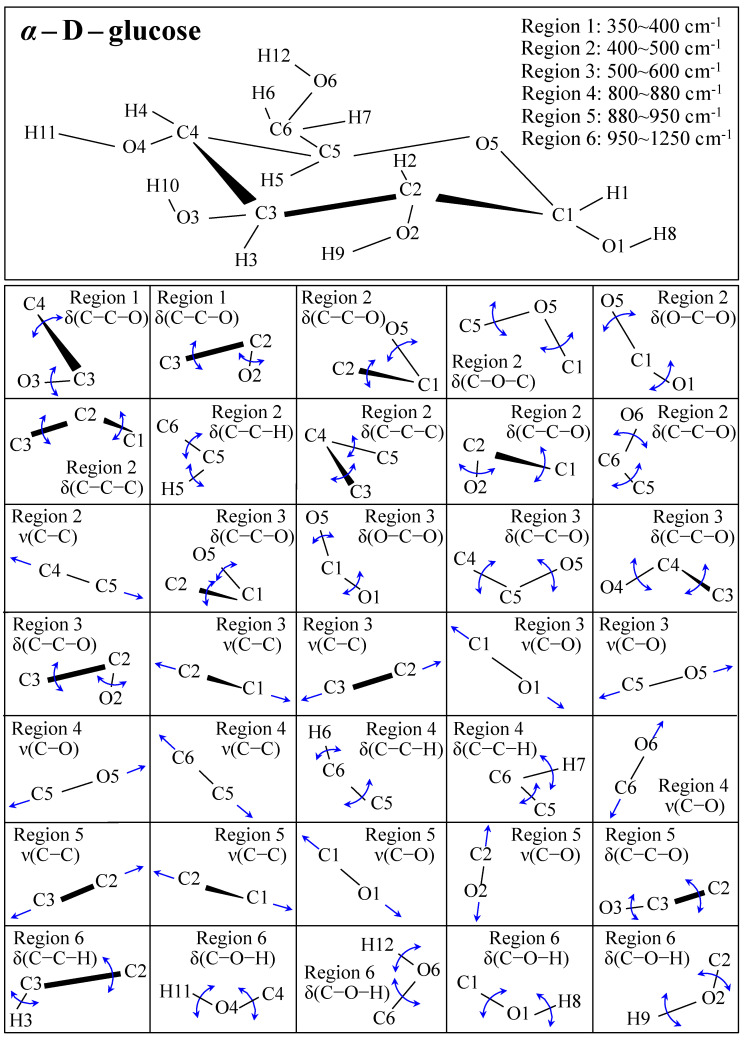
Schematic draft of the α–D–glucose with its principal stretching (ν) and bending (δ) vibrational modes and the related wavenumber intervals. Vibrational modes are schematically represented by means of blue arrows.

**Figure 4 foods-13-03068-f004:**
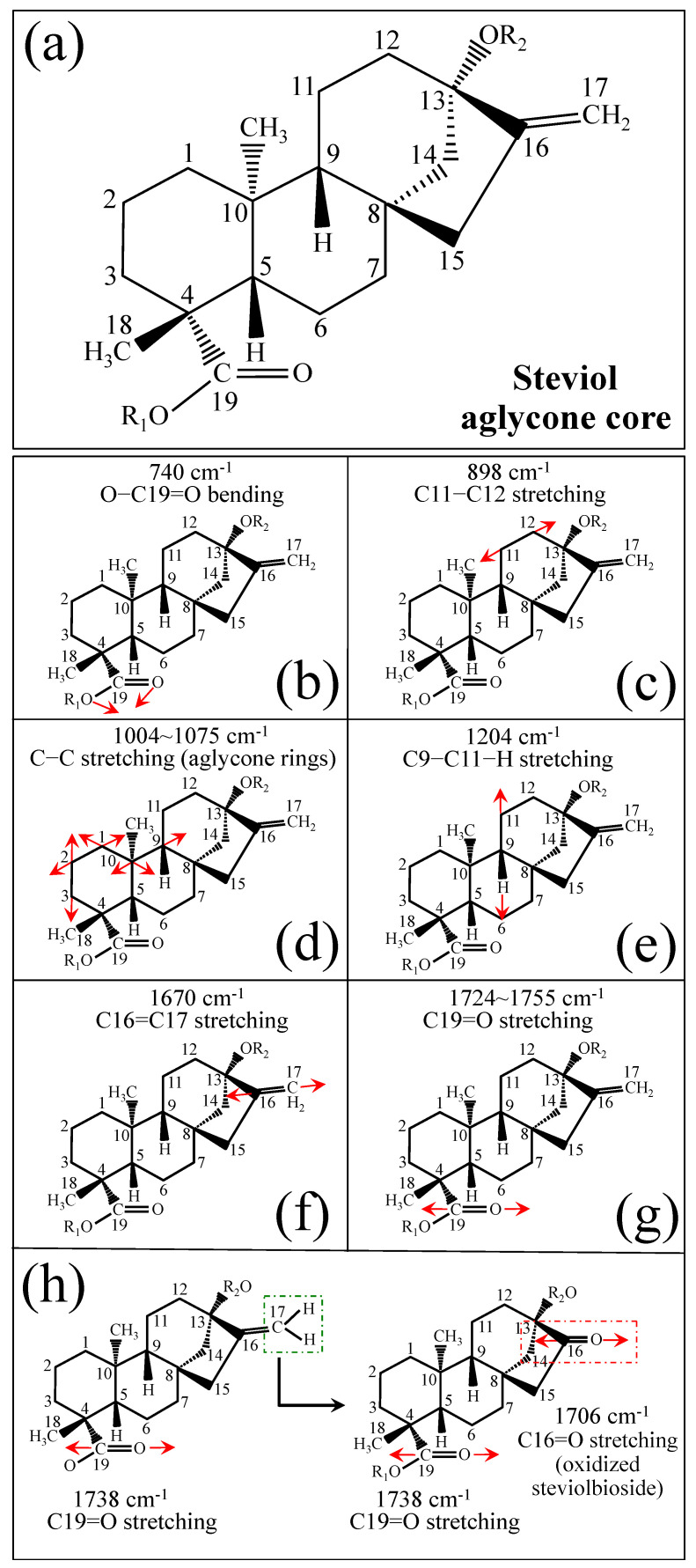
(**a**) Schematic draft of the aglycone core with (**b**–**h**) its principal stretching and bending vibrational modes and the related wavenumber intervals (cf. labels in inset). Red arrows schematically represents the vibrational modes of the molecules.

**Figure 5 foods-13-03068-f005:**
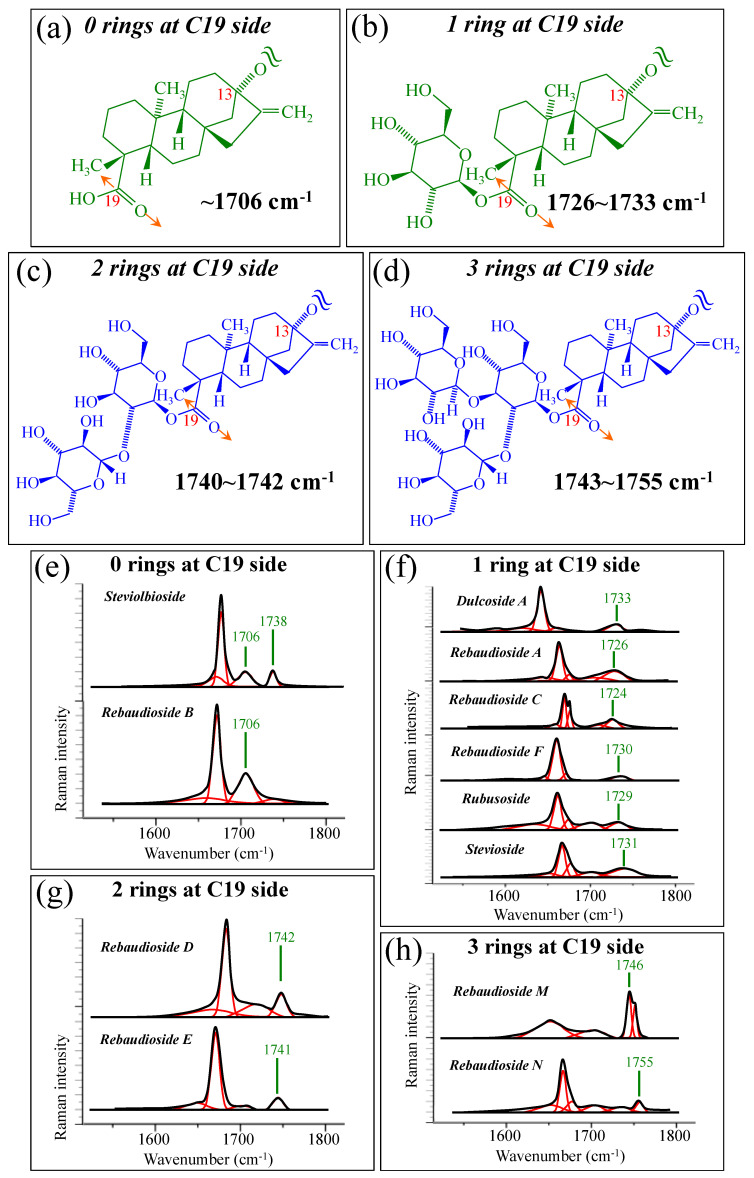
Schematic drafts of the C19 side of diterpene molecules with (**a**) 0 rings, (**b**) 1 ring, (**c**) 2 rings, and (**d**) 3 rings, and their related wavenumber intervals for carbonyl stretching vibration. In (**e**–**h**), high spectrally resolved 1600~1800 cm^−1^ regions are shown as detected for diterpene molecules with 0~3 glucose rings on their C19 side (cf. labels in in-set). Black and red lines represent spectra and deconvoluted sub-bands, respectively.

**Figure 6 foods-13-03068-f006:**
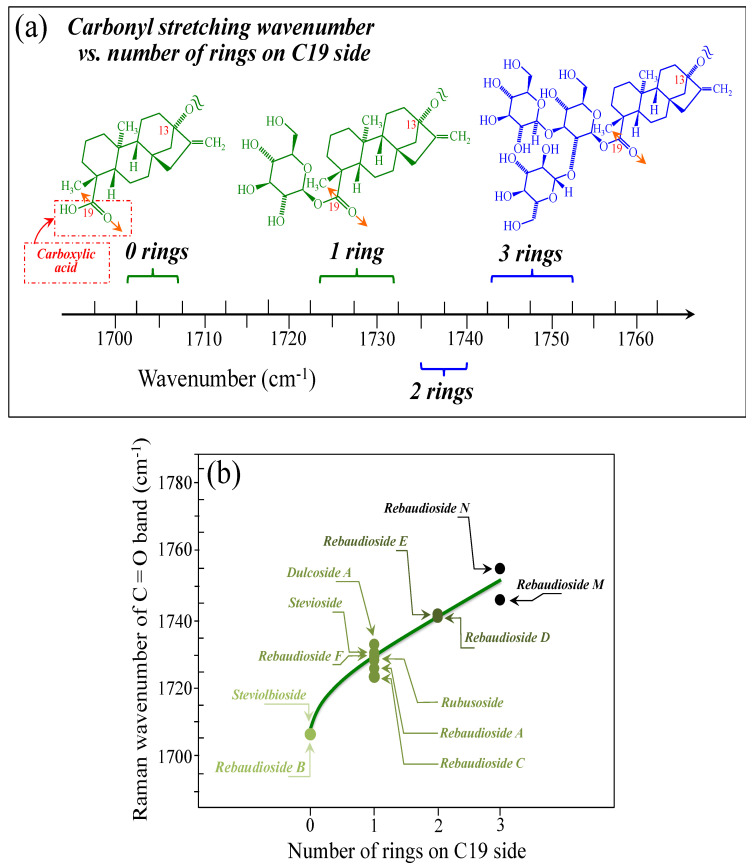
(**a**) Wavenumber intervals for the carbonyl stretching Raman vibration in diterpene molecules with glucose rings varying between 0 and 3 on their C19 side. In (**b**), a plot is shown of the carbonyl stretching wavenumber as a function of the number of rings on the C19 side of the diterpene molecule. Colors from light green to black emphasize the increase in the number of rings on C19 side.

**Figure 7 foods-13-03068-f007:**
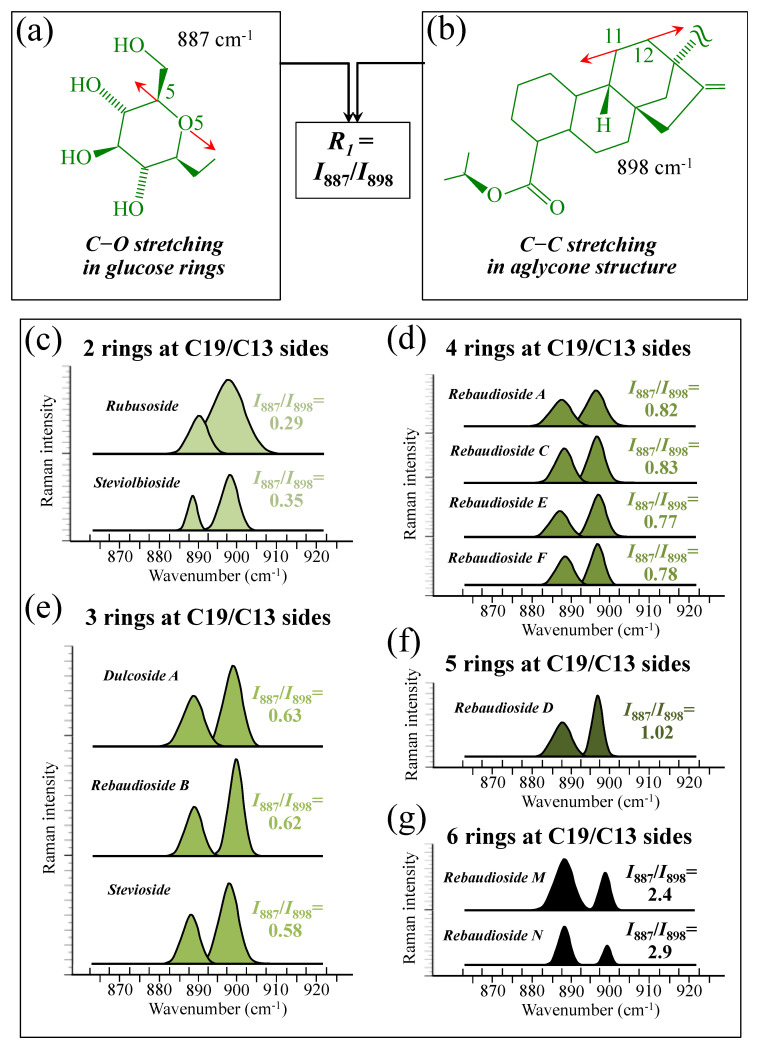
(**a**) Schematic drafts of the aglycone core of a diterpene molecule with its C11–C12 stretching vibration and the related wavenumber at 898 cm^−1^; (**b**) schematic draft of the glucose ring with its C5–O5 stretching mode and the related wavenumber at 887 cm^−1^ (red arrows indicate the selected vibrational modes). From the ratio between the (areal) relative intensities between the above two vibrational modes, the ratio, *R*_1_ = *I*_887_/*I*_898_, could be computed and used as a spectroscopic parameter to estimate the total number of glucose rings in the diterpene molecule. In (**c**–**g**), deconvoluted spectral components at 887 and 898 cm^−1^ are shown as detected for diterpene molecules with a cumulative number of 0~6 glucose rings on C13/C19 sides (cf. labels in inset). Colors from light green to black emphasize the increase in total number of rings on both C19 and C13 sides.

**Figure 8 foods-13-03068-f008:**
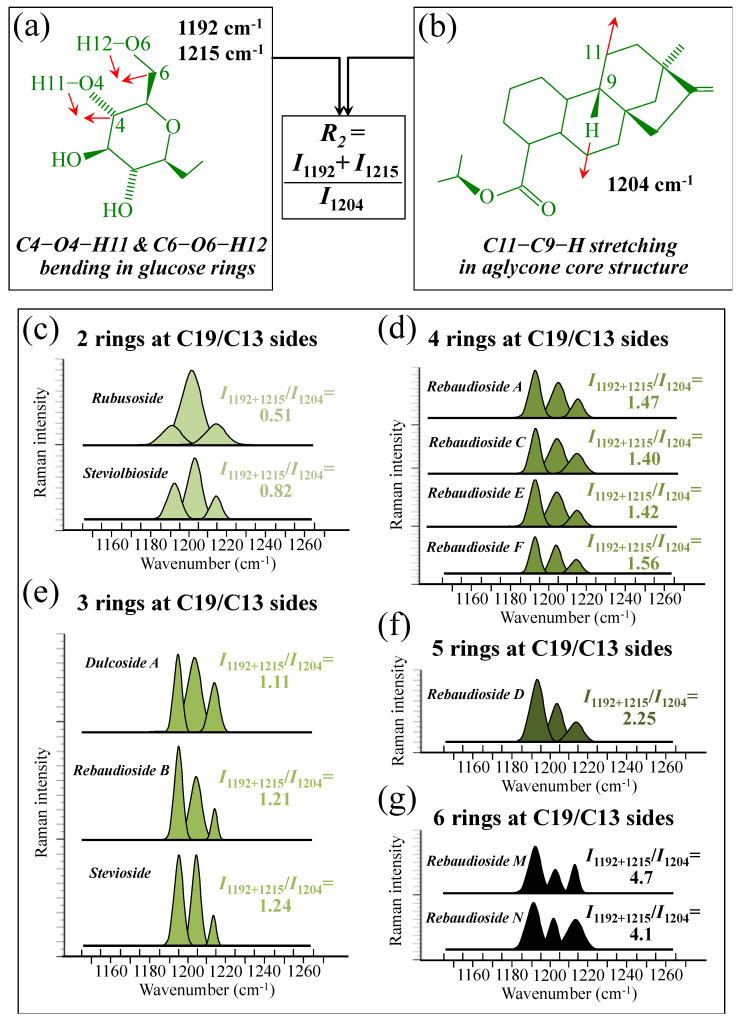
(**a**) Schematic draft of the glucose ring with its C6–O6–H12 and C4–O4–H11 bending modes and the related wavenumbers at 1192 and 1215 cm^−1^, respectively; and, (**b**) schematic drafts of the aglycone core of a diterpene molecule with its H–C9–C11 stretching vibration and the related wavenumber at 1204 cm^−1^ (red arrows indicate the selected vibrational modes). From the ratio between the (areal) relative intensities between the above vibrational modes, the ratio, *R*_2_ = *I*_1192+1215_/*I*_1204_, could be computed and used as a spectroscopic parameter to estimate the total number of glucose rings in the diterpene molecule. In (**c**–**g**), deconvoluted spectral components at 1192, 1204, and 1215 cm^−1^ are shown as detected for diterpene molecules with a cumulative number of 0~6 glucose rings on C13/C19 sides (cf. labels in inset). Colors from light green to black emphasize the increase in total number of rings on both C19 and C13 sides.

**Figure 9 foods-13-03068-f009:**
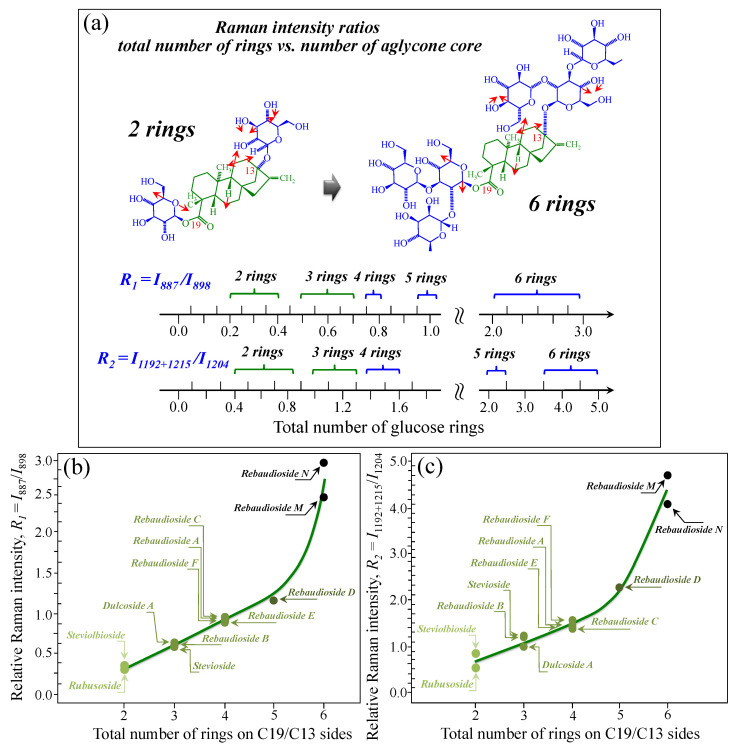
(**a**) Wavenumber intervals for the carbonyl stretching Raman vibration in diterpene molecules with a cumulative number of glucose rings varying between 2 and 6 on C13/C19 sides. In (**b**) and (**c**), plots are shown of the spectroscopic ratios, *R*_1_ and *R*_2_, obtained from selected stretching and bending spectral components (cf. Figure 7 and Figure 8) as a function of the cumulative number of glucose rings at C13/C19 sides. Colors from light green to black emphasize the increase in total number of rings on both C19 and C13 sides.

**Figure 10 foods-13-03068-f010:**
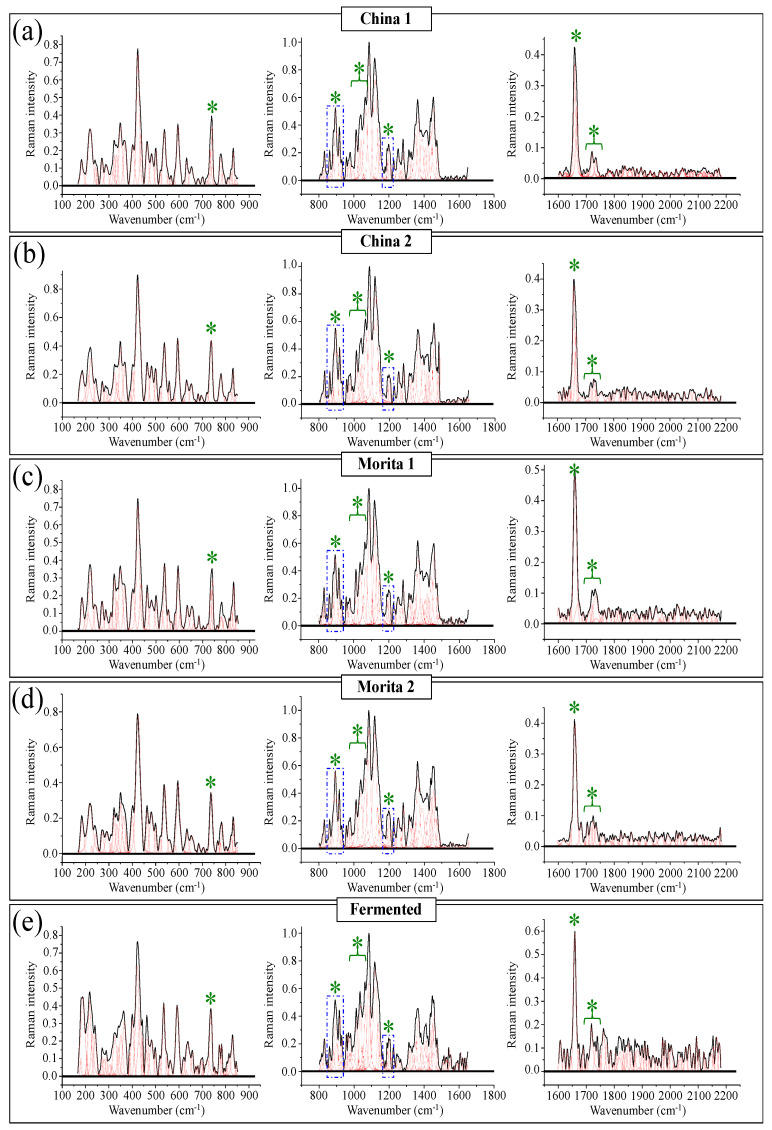
High-resolution Raman spectra of five different commercially available stevia products: (**a**) China 1, (**b**) China 2, (**c**) Morita 1, (**d**) Morita 2, and (**e**) Fermented. Each spectrum is divided into three wavenumber intervals at 150~850 cm^−1^, 800~1650 cm^−1^, and 1600~2150 cm^−1^. Black and red lines represent spectra and deconvoluted sub-bands, respectively. The meaning of green asterisks is given in the text.

**Figure 11 foods-13-03068-f011:**
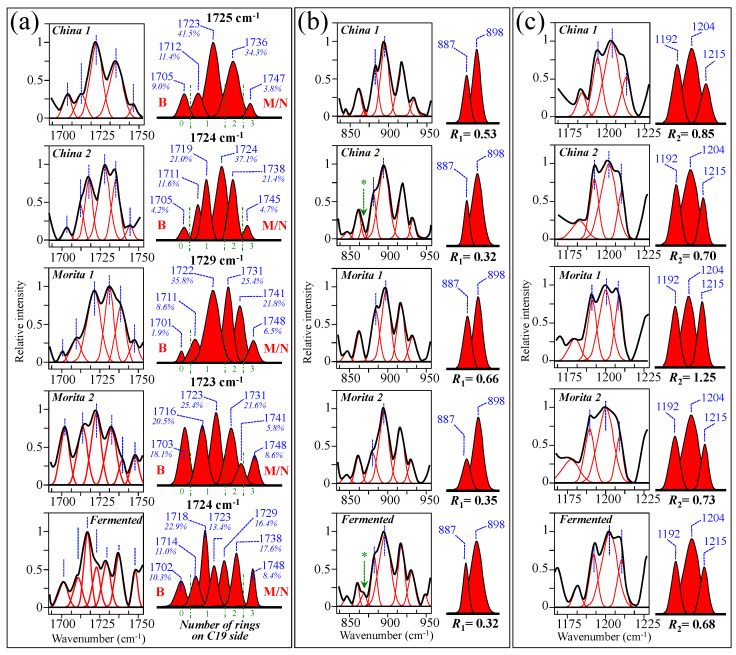
Spectroscopic analyses of five commercially available stevia products (cf. labels in inset) according to the spectroscopic parameters established in Figure 6 and Figure 9: (**a**) wavenumber of carbonyl stretching, (**b**) ratio *R*_1_ = *I*_887_/*I*_898_, and (**c**) ratio *R*_2_ = *I*_1192+1215_/*I*_1204_. Average values for the above three parameters are given in inset. The letters B and M/N in (**a**) refer to Rebaudiosides B (0 rings) and Rebaudioside M (3 rings)/N (3 rings), respectively. The percent fractions given in inset to (**a**) refer to the percentage of the area covered by each carbonyl sub-band over the total area of all sub-bands in the spectral zone of carbonyl stretching. The asterisks in (**b**) refer to a band belonging to an artificial sweetener added to adjust with both sweetness and bitter aftertaste in China 2 and Fermented samples.

**Figure 12 foods-13-03068-f012:**
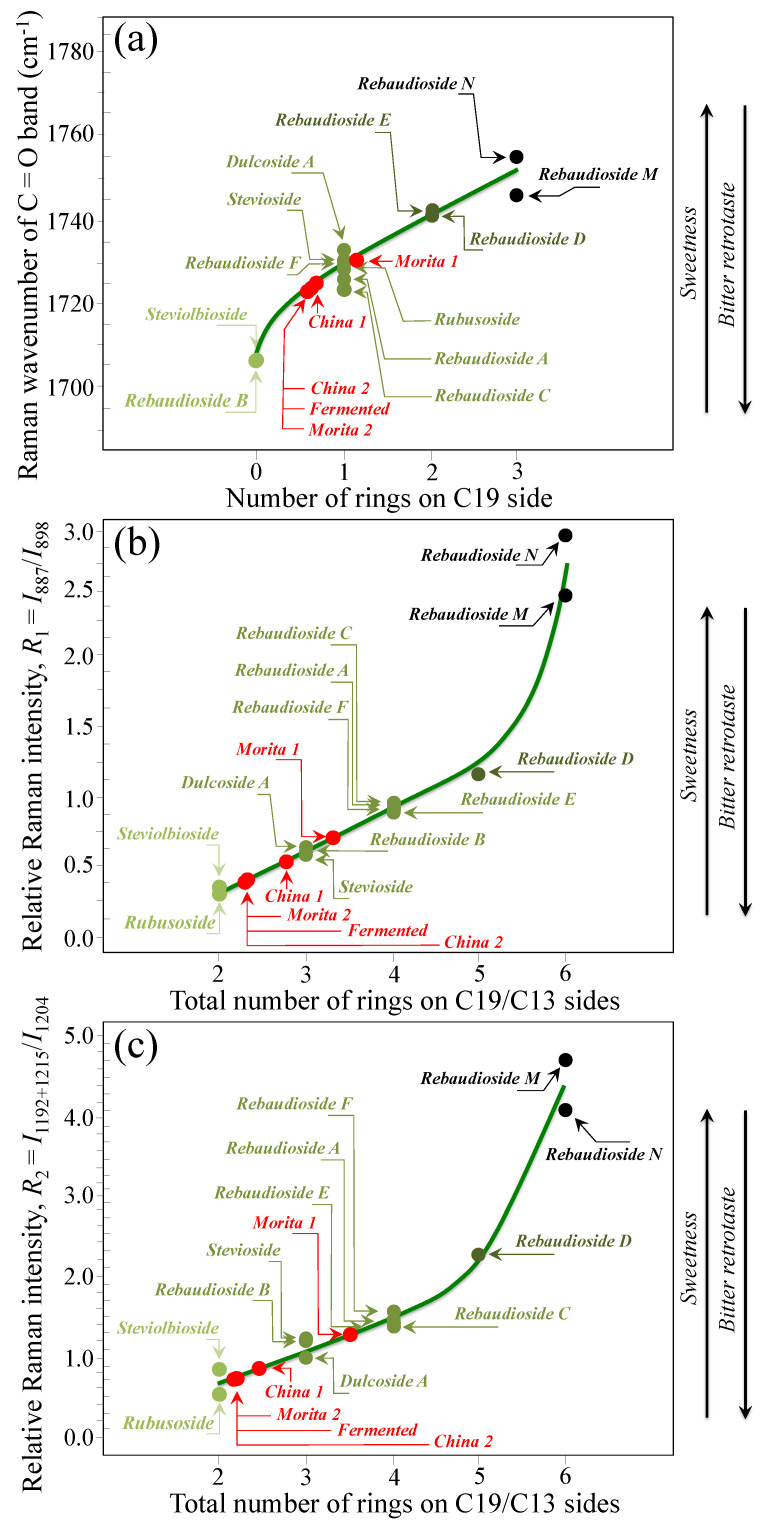
Matching of the (**a**) wavenumber of carbonyl stretching, (**b**) ratio *R*_1_ = *I*_887_/*I*_898_, and (**c**) ratio *R*_2_ = *I*_1192+1215_/*I*_1204_ average values, as determined for 5 commercially available stevia products, with the respective spectroscopic curves determined from Raman experiments on pure elementary compounds.

**Table 1 foods-13-03068-t001:** Chemical characteristics of the elementary diterpene molecules investigated in this study.

Commercial Name	Chemical Names	Chemical Formula	Molecular Weight	Rings on C19 Side	Rings on C13 Side
Dulcoside A	(4α)-13-{[2-O-(6-Deoxy-α-L -mannopyranosyl)-β-D -glucopyranosyl]oxy}-kaur-16-en -18-oic acid-β-D-glucopyranosyl ester	C_38_H_60_O_17_	788.87	1	2
Steviolbioside	(4α)-13-[(2-O-β-D-glucopyranosyl-β -D-glucopyranosyl)oxyl]-kaur-16-en -18-oic acid-β-D-glucopyranosyl ester; Steviosin	C_38_H_60_O_18_	804.87	1	2
Steviolbioside	(4α)-13-[(2-O-β-D-glucopyranosyl-β -D-glucopyranosyl)oxyl]-kaur-16-en -18-oic acid	C_32_H_50_O_13_	642.73	0	2
Rubusoside	(4α)-13-(β-D-glucopyranosyl)-kaur -16-en-18-oic acid, β-D -glucopyranosyl ester	C_32_H_50_O_13_	642.73	1	1
Rebaudioside A	(4α)-13-[(2-O-β-D-glucopyranosyl -(1→2)-O-[β-D-glucopyranosyl -(1→3)]-β-D-glucopyranosyl)oxy] -kaur-16-en-18-oic acid-β-D -glucopyranosyl ester; Stevioside α3; Rebiana	C_44_H_70_O_23_	967.01	1	3
Rebaudioside B	(4α)-13-[(O-β-D-glucopyranosyl -(1→2)-O-[β-D-glucopyranosyl -(1→3)]-β-D-glucopyranosyl)oxy] -kaur-16-en-18-oic acid; Stevioside A4	C_38_H_60_O_18_	804.87	0	3
Rebaudioside C	(4α)-13-[(O-6-Deoxy-α-L -mannopyranosyl-(1→2)-O-[β-D -glucopyranosyl-(1→3)]-β-D -glucopyranosyl)oxy]-kaur-16-en -18-oic acid-β-D-glucopyranosyl ester; Dulcoside B	C_44_H_70_O_22_	951.01	1	3
Rebaudioside D	(4α)-13-[(O-β-D-glucopyranosyl -(1→2)-O-[β-D-glucopyranosyl -(1→3)]-β-D-glucopyranosyl)oxy] -kaur-16-en-18-oic acid 2-O-β-D -glucopyranosyl ester	C_50_H_80_O_28_	1129.15	2	3
Rebaudioside E	(4α)-13-[(2-O-β-D-glucopyranosyl-β -D-glucopyranosyl)oxyl]-kaur-16-en -18-oic acid-2-O-β-D-glucopyranosyl -β-D-glucopyranosyl ester	C_44_H_70_O_23_	967.01	2	2
Rebaudioside F	(4α)-13-[(O-β-D-glucopyranosyl -(1→3)-O-[β-D-xlyopyranosyl -(1→2)]-β-D-glucopyranosyl)oxy] -kaur-16-en-18-oic acid -β-D -glucopyranosyl ester	C_43_H_68_O_22_	936.99	1	3
Rebaudioside M	(4α)-O-β-D-glucopyranosyl-(1→2)-O -[β-D -glucopyranosyl-(1→3)]-β-D -glucopyranosyl ester 13-[(O-β-D -glucopyranosyl-(1→2)-O-[β-D -glucopyranosyl-(1→3)]-β-D -glucopyranosyloxyl-kaur–16-en -18-oic acid; Rebaudioside X	C_56_H_90_O_33_	1291.29	3	3
Rebaudioside N	13-[(O-β-D-glucopyranosyl-(1→2)-O -[β-D-glucopyranosyl-(1→3)]-β-D -glucopyranosyl)oxy]-kaur-16 en -18-oic acid (4α)-O-6-deoxy-α-L -mannopyranosyl-(1→2)-O-[β-D -glucopyranosyl-(1→3)]-β-D -glucopyranosyl ester	C_56_H_90_O_32_	1275.29	3	3

**Table 2 foods-13-03068-t002:** Spectroscopic parameters and sensorial taste characteristics of the commercially available stevia products investigated in this study.

Product Name	Rings on C19	Total Rings 1 *	Total Rings 2 *	C13/C19 Ring Ratio	Taste Characteristics
Morita 1	1.15	3.31	3.31	1.88	Highest sweetness Fastest sweet perception Shortest bitter perception
Morita 2	0.58	2.33	2.33	3.02	Lowest sweetness Slowest sweet perception Longest bitter perception
China 1	0.69	2.78	2.78	3.03	Intermediate sweetness Slow sweet perception Long bitter perception
China 2	0.63	2.29	2.30	2.63	Lowest sweetness Slowest sweet perception Longest bitter perception
Fermented	0.61	2.30	2.31	2.77	Lowest sweetness Slowest sweet perception Longest bitter perception

1 *: using bands at 898 cm^−1^ (aglycone) and 887 cm^−1^ (glucose ring). 2 *: using bands at 1204 cm^−1^ (aglycone) and 1192/1215 cm^−1^ (glucose ring).

## Data Availability

The original contributions presented in the study are included in the article; further inquiries can be directed to the corresponding author.

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
