# Peer review of "Raman Spectroscopic Analysis of Steviol Glycosides: Spectral Database and Quality Control Algorithms"

_foods, 2024, doi:10.3390/foods13193068_

Round 1

Reviewer 1 Report

Comments and Suggestions for Authors

The manuscript has scientific merit, but it has deficiencies that need to be corrected, especially with regard to the detailed description of some of the procedures and methods. The main problem with the manuscript is the lack of description of the methods used to arrive at some of the results.

Line 28: Stevia rebaudiana must be italicized.

Lines 30-35: A "-" appears inappropriately in the text. Please revise.

121-136: There is "strange" text at the end of the introduction. Please remove it.

Line 352: How were the taste characteristics in Table II obtained? Were sensory tests performed? This needs to be clarified in the Materials and Methods.

Line 438: How were the algorithms determined? How were quantitative spectroscopic parameters determined? How were the curves developed? Regression? These questions need to be in the Materials and Methods.

490-513: What exactly do the percentages in brackets mean? How were they calculated? This needs to be clarified in the Materials and Methods.

520-521: How do you estimate the concentration of specific steviol glycosides in a sample? This is not clear. Please provide a more detailed explanation in the Materials and Methods.

The figures are low resolution. Please provide high-resolution (preferably vectorized) versions.

Author Response

We would like to thank both Reviewers for their constructive and insightful queries on our manuscript. They contributed to strengthen the contents and to achieve better clarity in the contents.

Reviewer 1

The manuscript has scientific merit, but it has deficiencies that need to be corrected, especially with regard to the detailed description of some of the procedures and methods. The main problem with the manuscript is the lack of description of the methods used to arrive at some of the results.

Line 28: Stevia rebaudiana must be italicized. CORRECTED

Lines 30-35: A "-" appears inappropriately in the text. Please revise. CORRECTED

121-136: There is "strange" text at the end of the introduction. Please remove it. REMOVED

Line 352: How were the taste characteristics in Table II obtained? Were sensory tests performed? This needs to be clarified in the Materials and Methods.

We clarified this point by adding Section 2.3 with the experimental details of the sensory test.

Line 438: How were the algorithms determined? How were quantitative spectroscopic parameters determined? How were the curves developed? Regression? These questions need to be in the Materials and Methods.

As requested by the Reviewer, we have now clarified this point. Section 2.2 now includes the requested details.

490-513: What exactly do the percentages in brackets mean? How were they calculated? This needs to be clarified in the Materials and Methods.

The meaning of the percent fractions shown in Fig. 11(a) has been added in the caption of Fig. 11. Additional explanations are given at the end of Section 4.2.

520-521: How do you estimate the concentration of specific steviol glycosides in a sample? This is not clear. Please provide a more detailed explanation in the Materials and Methods.

We added explanatory text in Section 3.3 specifying how the percent fractions given in inset to Fig. 11(a) are semi-quantitatively linked to the fractions of specific steviol glycosides in the overall product composition).

The figures are low resolution. Please provide high-resolution (preferably vectorized) versions.

We provided high-resolution figures in the revised manuscript.

Reviewer 2 Report

Comments and Suggestions for Authors

The manuscript discusses the sweetening power of diterpene glycosides, secondary metabolites from Stevia rebaudiana, which can be up to 450 times sweeter than sucrose. The research focuses on using Raman spectroscopy to develop algorithms for the quantitative analysis of stevia-based sweeteners, establishing a molecular structure library, and applying this to characterize the taste profiles of commercial products.

The manuscript as a whole is of good quality. The description of the methods and experimental design is thorough and clearly articulated. The overall presentation is well-written, the research design is appropriate, and all methods are adequately described. In my opinion, although the approach is not highly innovative due to several existing studies using Raman libraries in quality control, the original application to Stevia rebaudiana secondary metabolites and the overall experimental design make it scientifically interesting and potentially valuable for future quality control system development.

Some minor revisions should be considered before publication:

Throughout the manuscript, there are words truncated by hyphens (e.g., lines 35, 59, 61, etc.) and the “-1” in wavenumber should be superscript; these are likely formatting glitches that should be corrected.

In line 157: “Series of 10 spectra were collected (with a 50x optical lens) at different locations of each sample and then averaged to obtain a representative spectrum for each compound or product,” please provide additional details about the samples. Are the samples heterogeneous, or are the locations randomly selected, or are they the same for all samples?

Author Response

We would like to thank both Reviewers for their constructive and insightful queries on our manuscript. They contributed to strengthen the contents and to achieve better clarity in the contents.

Reviewer 2

The manuscript discusses the sweetening power of diterpene glycosides, secondary metabolites from Stevia rebaudiana, which can be up to 450 times sweeter than sucrose. The research focuses on using Raman spectroscopy to develop algorithms for the quantitative analysis of stevia-based sweeteners, establishing a molecular structure library, and applying this to characterize the taste profiles of commercial products. 

The manuscript as a whole is of good quality. The description of the methods and experimental design is thorough and clearly articulated. The overall presentation is well-written, the research design is appropriate, and all methods are adequately described. In my opinion, although the approach is not highly innovative due to several existing studies using Raman libraries in quality control, the original application to Stevia rebaudiana secondary metabolites and the overall experimental design make it scientifically interesting and potentially valuable for future quality control system development.

 Thank you very much for your kind words on our manuscript.

Some minor revisions should be considered before publication:

Throughout the manuscript, there are words truncated by hyphens (e.g., lines 35, 59, 61, etc.) and the “-1” in wavenumber should be superscript; these are likely formatting glitches that should be corrected.  CORRECTED

In line 157: “Series of 10 spectra were collected (with a 50x optical lens) at different locations of each sample and then averaged to obtain a representative spectrum for each compound or product,” please provide additional details about the samples. Are the samples heterogeneous, or are the locations randomly selected, or are they the same for all samples?

We have added this important detail in Section 2.2 of the revised manuscript.